# Heavy Metals Like Aluminum, Arsenic, Cadmium, Chromium, Copper, Iron, Lead, Manganese, Mercury, Nickel, and Zinc Polluting the Drinking Water: Their Individual Health Hazards

**DOI:** 10.3390/ijms262311656

**Published:** 2025-12-01

**Authors:** Rolf Teschke, Tran Dang Xuan

**Affiliations:** 1Department of Internal Medicine II, Division of Gastroenterology and Hepatology, Klinikum Hanau, 63450 Hanau, Germany; 2Academic Teaching Hospital of the Medical Faculty, Goethe University Frankfurt, 60629 Frankfurt am Main, Germany; 3Center for the Planetary Health and Innovation Science, The IDEC Institute, Hiroshima University, Higashi-Hiroshima 739-8529, Japan; tdxuan@hiroshima-u.ac.jp; 4Graduate School of Advanced Science and Engineering, Hiroshima University, Higashi-Hiroshima 739-8529, Japan; 5Graduate School of Innovation and Practice for Smart Society, Hiroshima University, Higashi-Hiroshima 739-8529, Japan

**Keywords:** heavy metals, polluted drinking water, health risks

## Abstract

Heavy metals (HMs) were originally formed in the universe long before human evolution and are now ubiquitous in the environment, where some HMs are good as essential elements for human health while others are not. The purpose of this analytical review is to provide an updated clinical overview on health risks attributable to drinking water containing specific HMs and to discuss new aspects of molecular steps leading to disrupted diseases. This approach was favored because the study cohorts were homogeneous, since exposed individuals lived in households where all members had access to the same drinking water of constant quality. Among the HMs under consideration, aluminum, arsenic, cadmium, chromium, copper, iron, lead, manganese, and mercury were detected in drinking water and represented a health risk if levels were above thresholds recommended by national and international regulatory authorities. For example, (1) aluminum increased the risk of dementia and Alzheimer’s disease; (2) arsenic was associated with the development of bladder cancer; (3) cadmium increased the no-carcinogenic, as well as the carcinogenic, health risk; (4) chromium was considered as a risk factor for liver and kidney injury, as well cancer development; (5) copper contributed to cognitive impairment in the aging population and Alzheimer’s disease; (6) iron increased the non-carcinogenic health risk; (7) lead impaired neurodevelopmental functions in children; (8) manganese increased the risk of attention-deficit hyperactivity disorder (ADHD); and (9) mercury was causally related to chronic kidney disease. In contrast, for nickel and zinc, no overt health risks have been reported, likely due to low levels in the drinking water, attributable to their low water solubility. Of note is the good news that some HMs represent essential elements for human health. In essence, many HMs were detected in drinking water and exerted non-carcinogenic or carcinogenetic health risks, requiring proactive management of national and international regulatory authorities.

## 1. Introduction

Heavy metals (HMs) are ubiquitarian inorganic chemicals in the environment that enter the human body via food and drinking water, with positive or negative effects on health conditions [1,2,3,4]. The general population commonly approaches the issues around HMs through their potential health risks and is less familiar with the positive properties of a few HMs, considered as essential elements to sustain human health. Among these are cobalt (Co), copper (Cu), iron (Fe), manganese (Mn), molybdenum (Mo), and zinc (Zn) [5,6,7,8,9,10]. For instance, cobalt is an essential molecule of cobalamin, or vitamin B_12_ [5], and copper is involved in various physiological processes like energy metabolism, antioxidant defense, blood clotting, neurotransmitter synthesis, and iron metabolism [6]. Iron is a key element for sustaining cell functions, contributing to survival, facilitating energy production, and helping cellular metabolism, and is important as a constituent of hemoglobin that carries oxygen via systemic circulation to all organs of the body [7]. Manganese is required for enzymes involved in various metabolic processes and, most importantly, is a structural component of the Mn superoxide dismutase [8]. Molybdenum is a chemical component of molybdenum-containing enzymes, including xanthine oxidase (XO), known for its involvement in purine catabolism but less appreciated for its capacity to produce reactive oxygen species (ROS) as a starting product of oxidative stress and subsequent organ dysfunction [9]. Finally, zinc is of relevance as a constituent of multiple enzymes and transcription factors [10]. It is obvious that these six HMs listed above belong primarily to the group of the good ones. However, for a few individuals with gene mutations, two heavy metals are crucial. This refers to the toxic accumulation of iron in genetically predisposed patients with iron-storing hemochromatosis [11]. The overload of copper in patients genetically predisposed to copper-storing Wilson disease is also critical [12]. This makes iron and copper intermediate HMs in terms of their essentiality and toxicity. There is much general data on HMs in the overall environment, as outlined in review articles by various authors, including our own group [1,2,3,4,5,6,7,8,9,10,11,12], but substantial information gaps exist in new data regarding the existence of HMs in the drinking water and health risks for humans consuming drinking water enriched with HMs.

The present review updated and analyzed the impact of selected environmental HMs on human health integrity, focusing on elements found in drinking water with amounts higher than allowed, as proposed by respective national and international regulatory authorities. The analysis attempts to provide new insights into how HMs disrupt cellular metabolic and molecular pathways required for human health integrity.

## 2. Literature Search

The PubMed database and Google Science were used for our analysis with the following terms: drinking water; surface water; heavy metals; health risks; hazard quotients; and environmental pollution by heavy metals. Only reports in the English language were considered; otherwise, there were no additional restrictions.

## 3. HMs and Human Health Risk Assessed for Exposures to Contaminated Drinking Water

A wealth of publications addressed the issue of health problems due to HMs, viewed among the general population as a major human threat [1,2,3,4]. Such occupational hazards were found among industry workers in heavy metal factories and were controversially discussed in relevant reports. However, this group of exposed persons is of a small size and not necessarily representative of a large population outside of direct exposures to industrial pollution by HMs. Current health risks due to HMs in workers are seemingly reducing as a consequence of protective measures in the work place to minimize exposures. Discussions on occupational health effects may be hampered by confounding conditions such as the inclusion of only the male gender, overlooking other pre-existing diseases, deleterious effects of smoking or alcohol abuse, exposure to a mix of HMs, and routes of exposure by dermal contact or via inhalation. Considering these points critically, cohorts consisting of individuals confronted by occupational sequalae may not be the ideal study cohorts. Instead, cohorts of households exposed to drinking water with HMs as pollutants may be a better group to study.

Members of a household represent an ideal homogeneous cohort regarding the consumption of drinking water contaminated with HMs, which may pose health risks. Among the HMs most commonly detected in drinking water are aluminum, arsenic, cadmium, chromium, copper, iron, lead, manganese, mercury, nickel, and zinc, although the levels of pollutants and health risks vary substantially. The consumption of these eleven HMs was analyzed for their potential health risks (Table 1) [13,14,15,16,17,18,19,20,21,22,23,24,25,26,27,28,29,30,31,32,33,34,35,36,37,38,39,40,41,42,43,44,45,46,47,48,49,50,51,52,53,54,55,56,57,58,59,60,61,62,63,64,65,66,67,68,69,70,71,72,73,74,75,76,77,78,79,80].

All eleven HMs under consideration were detectable in drinking water; nine of these represented health risks to exposed people (Table 1). Contrastingly, levels of nickel and zinc in drinking water were low, attributed to low water solubility, and represented no human health risk [76,80]. Assessing health risks among the cohorts was achieved through evaluating obvious ailments of exposed individuals or by correlating increased HM levels in drinking water with regulatory recommendations for the maximal levels allowed (Table 1). In general, risks increased with the amount of ingested HMs and duration of the intake and were dependent on the valency of the HMs. On theoretical grounds, the risk of the ten HMs may be augmented if combined with the same HM originating from other sources or if other HMs with potential health risks were consumed.

With its minerals such as calcium, chloride, fluoride, iodine, magnesium, phosphorus, sodium, and potassium [81], water is an essential product to sustain human health. Long before the metal industry contributed its HMs to the environment, including water resources, drinking water containing heavy metals was a reality, since humans are on Earth because HMs originated from natural sources after arriving on our globe from the universe, where they were formed [4]. During mankind’s evolution, humans developed mechanisms to cope with the low amounts of HMs by incorporating these partially into important enzymes, because humans cannot synthesize essential HMs.

## 4. Mechanistic Aspects of Injury Caused by Selected HMs

HMs are responsible for a wide range of mechanistic steps leading to health issues via disrupting metabolic pathways required for human health integrity. The pathogenetic variability reflects differences in molecular characteristics, metabolic pathways, and target organs (Table 2) [2,3,4,11,12,22,23,82,83,84,85,86,87,88,89,90,91].

For many HMs found in drinking waters across several countries throughout the world (Table 1), specific molecular details are described regarding how HMs may initiate the toxic effects (Table 2) leading to health problems (Table 1). The injury is to a large part causally related to the generation of the reactive oxygen species (ROS), responsible for cellular oxidative stress in subcellular organelles like mitochondria or the endoplasmic reticulum (Table 2). This eventually leads to cell death through disrupting essential metabolic pathways.

## 5. HMs from Drinking Water to Human Gut, Liver, and Organs Systems

### 5.1. Gastrointestinal Absorption

The rate of intestinal absorption is variable among the HMs, with low rates that reduce health risks, but estimates of gastrointestinal absorption remain challenging [92,93]. HM absorption in exposed humans also depends on the chemical form of the HMs, contents of food in the gastrointestinal tract, and the genotype in some patients [92,94]. In addition, the bioavailability and toxicity of HMs is influenced by their intestinal permeability [93] and is modified by alterations to the gut microbiome [93,94,95].

### 5.2. Intestinal Microbiome and Dysbiosis

The human gut microbiome allows for the maintenance of intestinal homeostasis [94,95]. However, HM exposure disrupts gut microbiota, perturbations that cause intestinal dysbiosis. This impairs protective properties of the gut microbiome on HM toxicity through altering intestinal pH, disrupting the oxidative balance, inhibiting activities of detoxification enzymes, or modifying enzyme proteins involved in HM degradation [95]. Despite various inhibitory effects of dysbiosis, ingested HMs are absorbed by the intestinal cells via passive diffusion, intracellular and extracellular ligands, or various other transcellular and paracellular pathways [92,93]. The subsequent transfer of the HMs from the enterocytes to the human organism proceeds through the blood of the tributary portal vein, which brings the HMs to the liver cells, where they are internalized by specific transporter systems present in the plasma membrane. The pivotal contribution of the liver in handling HMs is evidenced by the clinical observation that HMs may cause toxic liver injury [2,3,4].

### 5.3. HMs and Disrupted Homeostasis

The liver is flooded by the portal vein blood that contains the absorbed HMs. In the liver, HMs may undergo degradation, biliary excretion, or organ systems, where transporter systems allow for HM incorporation in cells that are then targets for the injury and disease [2,3,4,96,97,98]. The harmful effects of HMs on health integrity are to a large extent ascribed to their interaction with mechanisms related to antioxidant defense, with a focus on glutathione or sulfhydryl groups contained in antioxidant enzymes such as superoxide dismutase, catalase, glutathione peroxidase, and glutathione reductase [96].

### 5.4. Critical Role of ROS and Vicious Cycles

HMs are toxic as their ions with variable valences, which may initiate cellular oxidative stress in the context of ROS generation, leading to disruption of cellular homeostasis. These processes have been well described for iron in the iron-storing disorder known as hemochromatosis [11] and copper in the copper-storing disorder known as Wilson disease [12]. The injurious events proceed by the Haber Weiss or Fenton reactions (Table 3).

The Haber Weiss and Fenton reactions are shown for copper and iron as examples (Table 3); both elements can be replaced by other HM ions if the appropriate valences are provided. There is also the theoretical option that during injurious intracellular processes, iron is released from iron-containing enzymes, leading to increased rates of ROS generation and health risks.

Some additional comments are warranted to better understand the role of the Haber Weiss and Fenton reactions. There is much direct evidence that copper produces ROS with hydroxyl radicals as its main toxic product, perfectly ascertained by studies using electron paramagnetic resonance (EPR) spectroscopy measurements and electron spin resonance (ESR) studies [12,99]. In analogy to the equations above (Table 3), even if copper or iron are replaced by other HM ions, the injurious processes are maintained and may even be augmented in the sense of a vicious circle, as shown previously for the combination of copper ions with iron ions [12]. These two HMs participate in the production of ROS via the Haber Weiss and Fenton reactions, providing ROS with identical radical types as a net result (Table 3). Overall conditions may become even worse as both metals now can each create their own vicious cycle (Table 3) [12]: Cu^2+^ is reduced to Cu^1+^, which in turn is converted back to Cu^2+^. Concomitantly, Fe^3+^ is oxidized to Fe^2+^, which in turn is converted back to Fe^3+^. Both vicious cycles obviously contribute to liver injury perpetuation through a non-stop mechanism based on irreversible chain reactions.

## 6. HM Issues Outside the Drinking Water

The current analysis of health issues is limited to HMs in drinking water (Table 1 and Table 2), but health risks may be augmented by other settings of environmental pollution by HMs.

### 6.1. Herbal Medicines

Herbal medicines occasionally contain HMs such as antimony, arsenic, cadmium, chromium, copper, iron, manganese, mercury, nickel, selenium, thallium, tin, and zinc with their potential health risks [100]. Most critical is the Ayurvedic medicine, because its use causes herb-induced liver injury (HILI) [101,102,103] with verified diagnosis [101] using the updated Roussel Uclaf Causality Assessment Method (RUCAM) [104], which provided a probable causality grading for the herbal Ayurvedic products [100], confirmed by similar reports with HILI cases [101,102] assessed by the original RUCAM [105]. However, it remained unclear whether the toxicity was due to the herbal mixtures per se or to the arsenic or mercury detected in the herbal Ayurveda mixtures, which caused a high death rate among the study cohort [101].

### 6.2. Edible Plants and Other Food

Edible plants like vegetables can easily be contaminated by HMs via polluted soil present around municipal solid waste landfills [106]. Chemical analysis of plants nearby a landfill showed elevated levels of chromium, copper, lead, and zinc in the roots and stems of *Ipomea aquatica*, a highly appreciated vegetable for consumption of its leaves and stems by local villagers in Vietnam, in amounts much higher than the World Health Organization (WHO) standards, which represents risks to human health via the food chain [106].

### 6.3. Minamata Disease

A deleterious HM effect was reported in Japan in 1956. Mercury was detected in fish and shellfish, and their ingestion led to Minamata disease caused by methylmercury poisoning if humans ingested the contaminated fish and shellfish [107]. Toxicological analyses revealed that methylmercury has been removed as wastewater by a chemical plant in Minamata City, located in the south-west region of Kyushu Island. Various marine products collected in Minamata Bay displayed high levels of contaminating mercury (5.61 to 35.7 ppm). Concomitantly, mercury content in the hair of patients, their family, and inhabitants of the Shiranui Sea coastline was also detected at high levels (maximum 705 ppm), substantiating the causative role of mercury in the development of Minamata disease. Typical clinical symptoms in patient experiencing Minamata disease included sensory disturbances of the glove and stocking type, ataxia, dysarthria, constriction of the visual field, auditory disturbances, and tremor [49,107]. Even worse, fetuses in the womb suffered from the congenital Minamata disease with microcephaly, extensive cerebral damage, and symptoms similar to those seen in cerebral palsy, if poisoned by methylmercury when their mothers ingested contaminated marine products [107,108]. For the past 36 years, 1043 patients have died [107]. This is one of the most tragic regrettable outcomes of a pollution disease by HM exposure, with congenital malformations at birth, health impairment in adults, and high death rates [107,108]. Additional information about Minamata disease was provided in a more recent paper [90].

### 6.4. Occupational Exposure

Occupational exposure to HMs can occur at many work places across various professions and represents health risks if HM levels exceed safe limits [109,110,111]. HMs still affect workers via inhalation [111] or dermal contact [112], although attempted protective measures by the metal industry or other employers have been mostly effective, as evidenced by the mean concentrations of HMs in occupational environments being lower than the US National Institute for Occupational Safety and Health (NIOSH) thresholds [109].

Inhalation of HMs was viewed as a household and occupational health risk [109,113]. Surprisingly, analyses of HM concentrations revealed that levels in most of the tested households exceeded the WHO guideline values, indicating moderate pollution and dominant anthropogenic emission sources of HMs [113]. In the analyzed schools, universities, and offices, low-to-moderate levels of air pollution with HMs were recorded. Conversely, in commercial environments of certain industrial production areas, high grades of air pollution prevailed. The non-carcinogenic risk due to HM inhalation exceeded the acceptable level of 1.0 in households, cafes, hospitals, restaurants, and metros. The carcinogenic risk was variable: for arsenic and chromium in households, for cadmium, chromium nickel, arsenic, and cobalt in educational settings, for cadmium, cobalt, chromium, and lead in offices and commercial environments, and for nickel in metros, exceeded the acceptable level of 1 × 10^−4^ [113]. For HM inhalation, the carcinogenic risk was higher indoors than outdoors.

Occupational dermal HM exposure has been studied in employees of the batik industry in Indonesia [112]. The industry uses synthetic and traditional dyes that contain HMs. The X-Ray Fluorescence (XRF) method was applied to patch filters that were attached to the skin of workers during the sampling period to be analyzed for the dermal uptake of HMs. The patch filters detected several HMs including aluminum, copper, iron, lead, nickel, and zinc [5]. The highest values of the hazard quotient (HQ) were found for copper, iron, manganese, and zinc in workers engaged in the dyeing process when compared to those in other processing stages.

## 7. Current and Future Challenges

There is an urgent need to harmonize and globalize efforts among countries, regions, regulatory authorities, and institutes, including the United States, UN, and the WHO, to strengthen new policies and approaches aiming to solve the issue of drinking water contaminated with HMs. In particular, proactive management should focus on precise biomonitoring for HM detection, prevention of water contamination, and measures to remove HMs from drinking water and the environment. Artificial Intelligence (AI) and machine learning (ML) can assist in the identification and further remediation of water and wastewater from HMs [114].

## 8. Conclusions

Drinking water contaminated with HMs is the predominant source of HMs detectable in humans. While some of these are essential elements with benefits for human health, others represent human risk factors. The human organism cannot synthesize HMs; their presence in the body reflects uptake from outside sources. There is general agreement that the non-carcinogenic and carcinogenic health risks caused by HMs are due to the generation of ROS, with processes carried out by the Haber Weiss and the Fenton reactions. ROS in turn provides reactive intermediates such as singlet radical ^1^O_2_, superoxide radical HO•_2_, hydrogen peroxide H_2_O_2_, hydroxyl radical HO•, alkoxyl radical RO•, and peroxyl radical ROO•. These toxic radicals injure structural membranes of subcellular organelles like mitochondria or the endoplasmic reticulum and thereby initiate cell injury that may result in programed cell death. In addition, the radicals may covalently combine with DNA, a precursor process that may eventually lead to cancer. As a consequence, the consumption of drinking water contaminated with high levels of HMs may represent health risks, resulting in substantial morbidity and mortality rates all over the world. To reduce human health risks, there is a need to strengthen new regulatory policies at the national and international level, with AI assisting in biomonitoring.

## Figures and Tables

**Table 1 ijms-26-11656-t001:** Heavy metals in drinking water and their health risks to exposed humans.

Heavy Metal	Published Health Risks to Exposed Humans	References
Aluminum (Al)	Aluminum is a common pollutant of drinking water and a cause of health problems. In a prospective study from France, the relative risk of dementia, adjusted for potential confounding variables, was 2.03 for individuals exposed to aluminum concentrations greater than 0.1 mg/L in drinking water. Human health risk of aluminum in drinking water treated with aluminum-based coagulants in a rural area was a theoretical concern but the calculated Hazard Index for children and adults showed no health risk, with values < 1.0.	Rondeau, 2000 [13] Krupińska, 2020 [14] Cutipa-Díaz, 2024 [15] García-Ávila, 2025 [16]
Arsenic (As)	Arsenic in higher amounts detected in drinking water plays a significant role as health hazard and was evaluated in a systemic review with a focus on epidemiology issues during a period of 30 years. An overt association between arsenic content in drinking water and the occurrence of bladder carcinoma was revealed in 28 studies. Based on the evaluated meta-analyses, a predicted risk of bladder cancer incidence of 5.8 was found for drinking water with arsenic levels of 150 μg/L. For bladder cancer, mortality rates were 30% greater at arsenic levels of 150 ug/L than those at 10 μg/L. Arsenic reduction in drinking water reduces lethality due to chronic diseases. There is an overall call for additional regulations and efficient public health approaches to reduce arsenic contamination in drinking water to ensure public health.	Saint-Jacques, 2014 [17] Yang, 2020 [18] Abtahi, 2023 [19] Frisbie, 2022 [20] Kumar, 2025 [21] Wu, 2025 [22]
Cadmium (Cd)	Cadmium, as contaminant of groundwater used as drinking water in rural regions of Iran, was assessed in individuals drinking this kind of water for risks affecting their health. The cadmium amounts detected in the groundwater of the evaluated regions ranged from 0.087 to 14.32 μg/L and from 0.417 to 18.36 μg/L. The health risk for cadmium contamination, expressed as quotient among children and infants, was >1.0 × 10^−4^. Similarly, the carcinogenic risk of cadmium in drinking water for adults, children, and infants was higher than the safe limit of 1.0 × 10^−4^, supporting the view of increased cancer risk among the population drinking water contaminated with cadmium.	Qasemi, 2019 [23] Ahmed, 2020 [24] Decharat, 2023 [25] Simran, 2025 [26]
Chromium (Cr)	Chromium as Cr^6+^ is a potentially toxic metal detected in drinking water and groundwater of either natural or anthropogenic origin and represents a serious health problem for EU countries. Risks to human health include hepatic and renal injury, internal hemorrhage, DNA damage, and evolution of cancer, with the risks strongly depending on the duration of chromium exposure and the amount of chromium ingested.	Tumulo, 2020 [27] Whitaker, 2020 [28] Chandio, 2021 [29] Georgaki, 2023 [30] Xie, 2024 [31] Paydar, 2025 [32]
Copper (Cu)	Copper is a common heavy metal found in drinking water provided via copper plumbing in households of developed countries. Due to its known systemic toxic potential, copper contributed to reduced cognitive functions in the elderly and specifically in patients experiencing Alzheimer disease. Inorganic copper in drinking water does not significantly enter the liver. Instead, it remains in the blood and supplements the copper pool, where it is detected again as free copper. Via systemic circulation, the free copper enters many organs, including the brain, resulting in neuro-degenerative disruptions.	Fitzgerald, 1998 [33] Eife, 1999 [34] Araya, 2004 [35] Brewer, 2009 [36] Gomes, 2019 [37] Manne, 2022 [38] Montagnino, 2022 [39]
Iron (Fe)	Iron consumed through drinking water was evaluated for human health risks by applying hazard quotients (HQ) for adults and children. As a result, the non-carcinogenic health risk due to ingestion of iron was up to 1.5 for adults, but it was substantially lower among children, for whom no health risks were found. The assessment method proposed by the United States Environmental Protection Agency (US EPA) was applied.	Ghosh, 2020 [40] Haque, 2021 [41] Sharma, 2021 [42] Hu, 2024 [43] Rahman, 2024 [44]
Lead (Pb)	Lead can easily reach drinking water sources via lead-based service lines, especially through lead-containing plumbing. It is common knowledge that even low amounts of lead can impair neurodevelopmental functions. In addition and even worse, under conditions of already small amounts in drinking water an association between lead levels of the drinking water with those of the blood was detected, as evidenced by studies focusing on analysis among populations and based on toxicokinetic approaches. As expected, several methods are available that help reduce the lead level in drinking water.	Watt, 2000 [45] Payne, 2008 [46] Hanna-Attisha, 2016 [47] Rosen, 2017 [48] Levallois, 2018 [49] Fawkes, 2021 [50] Jarvis, 2021 [51] Bauza, 2023 [52] Cuomo, 2023 [53] Decharat, 2023 [25]
Manganese (Mn)	Manganese detectable in drinking water may exert toxicity, leading to neurodevelopmental disorders. According to a nationwide population-based registry study from Denmark, higher manganese levels in drinking water were causally related with a higher risk of the attention-deficit hyperactivity disorder (ADHD) inattentive subtype, but not of the ADHD combined subtype. After adjusting for age and birth year, females exposed to high levels of manganese, > 100 μg/L, in drinking water at least once during their first 5 years of life had a hazard ratio (HR) for ADHD inattentive subtype of 1.51. In contrast, the corresponding value was 1.20 in males when compared with same-sex individuals exposed to <5 μg/L drinking water.	Hafeman, 2007 [54] Ljung, 2007 [55] Iyare, 2019 [56] Kullar, 2019 [57] Rodrigues, 2019 [58] Schullehner, 2020 [59] Rahman, 2021 [60] Kumar, 2024 [61] Browning, 2025 [62] Rahman, 2025 [63]
Mercury (Hg)	Mercury detected in the domestic water of the delta region of Egypt was causally related to chronic kidney disorder due to accumulation of mercury in the kidneys, especially in the proximal tubule cells. In addition, mean levels of mercury in water and urine samples of inhabitants of the delta region exceeded standard references, denoting high exposure to mercury. Of clinical importance is the Minamata disease caused by mercury, as discussed in detail below in the text.	Abdeldayem, 2022 [64] Deziel, 2024 [65] Pant, 2024 [66] Charkiewicz, 2025 [67] Kayani, 2025 [68] Xu, 2025 [69]
Nickel (Ni)	Nickel in drinking water has been a substance of priority of the European Water Framework Directive for a long period, since 2008, originally viewed as a European Union-wide risk regarding surface waters. However, it turned out based on studies carried out since 2008, that nickel exerts very low risks when present in contaminated drinking water derived from surface waters. According to these new data, major risks to human health are not to be expected. Carbonyl, in its most powerful form, is poorly soluble in water, which may prevent its broad occurrence in water.	Payment, 2003 [70] Alam, 2008 [71] Haber, 2000 [72] Genchi, 2020 [73] Wang, 2020 [74] Adhikari, 2022 [75] Peters, 2022 [76] Salehi, 2024 [77] Simran, 2025 [78]
Zinc (Zn)	Zinc is detectable in drinking water in some parts of China, which is known as one of the largest producers and consumers in the world. Surprisingly, the analysis revealed low zinc amounts in drinking water across various settings, with values of non-cancer risks estimated to be 0.13 × 10^−12^ for Zn. The low zinc level in the drinking water associated with a negligible impact on human health may be due to its low water solubility.	Plum, 2010 [79] Huang, 2015 [80]

**Table 2 ijms-26-11656-t002:** Suggested mechanistic steps leading to toxicity by HMs of drinking water.

Heavy Metal	Mechanistic Steps Leading to Health Risks	References
Aluminum (Al)	Aluminum impairs human health causally due to disturbed cellular homeostasis through disruption of essential metabolic pathways or inhibition of important cellular enzymes activities. This modifies, among other functions, the synthesis structural and soluble proteins, reduces the working potential of nucleic acids, negatively affects cell membrane permeability, prevents repairing processes of the DNA, destabilizes DNA organization, inhibits protein phosphatase 2A activity, augments the production of reactive oxygen species (ROS) as a precursor of oxidative stress, impairs the activity of antioxidant enzymes, modifies cellular iron homeostasis, and influences the nuclear factor kappa-light-chain-enhancer of activated B cells (NF-kB), Tumor suppressor Protein-53 (p53), and c-Jun-N-terminal kinases (JNKs) pathway leading to apoptosis.	Teschke, 2022 [2] Teschke, 2022 [3] Rahimzadeh, 2022 [82]
Arsenic (As)	Arsenic’s toxicity is attributed to emerging cellular oxidative stress triggered by ROS, which are generated during the reduction process of molecular oxygen under inclusion of the superoxide radical anion, hydrogen peroxide, hydroxyl radical, hydroperoxyl radical, and peroxyl radical. As soon as an imbalance occurs between the high production rate of ROS and ROS consumption to be used for physiological or detoxification processes, ROS will accumulate and initiate oxidative stress. Due to the surplus production of ROS, different signaling pathways become altered, leading to oxidative modifications of biomolecules and causing concomitant loss in function of proteins, organelle damage, and even death of cells.	Teschke, 2024 [4] Ganie, 2023 [83]
Cadmium (Cd)	Cadmium lacks mechanisms of elimination, resulting in its continuous accumulation in the body throughout the lifespan. At the molecular level, cadmium triggers toxicity via cellular mitochondrial or endoplasmic reticulum-based oxidative stress, disruption of calcium signaling pathways, interference with cellular signaling processes, and epigenetic modifications. On mechanistic grounds, cadmium interacts with cellular components such as mitochondria and DNA, disrupting the physiological cellular homeostasis and the balance between oxidants and antioxidants, finally leading to cellular damage and apoptosis. Additionally, cadmium interferes with signaling pathways like Mitogen-Activated Protein Kinase (MAPK), NF-κB, and p53 pathways.	Teschke, 2024 [4] Qu, 2024 [84]
Chromium (Cr)	Chromium, especially as Cr^6+^, augments cellular mitochondrial and endoplasmic reticulum-based oxidative stress, causes chromosome breaks, and facilitates DNA adduct formation. The liver injury caused by chromium is primarily attributable to the toxic ROS produced in the course of cellular oxidative stress. These processes, disturbing cellular homeostasis, are responsible for the apoptosis of liver cells. Supporting evidence for the role of oxidative stress and lipid peroxidation by chromium as causatives in the liver injury can be retrieved from the high malondialdehyde levels detected under these conditions. Moreover, the hepatic content of antioxidant glutathione, nonprotein thiol, and vitamin C was decreased. Concomitantly, lower activities of antioxidant enzymes like glutathione peroxidase and superoxide dismutase were observed. There is also good evidence that chromium initiates apoptosis and inflammation by processes that inhibit the deacetylation of SIRT1, which stands for sirtuin or silent mating-type information regulation 2 homolog as a member of a protein family involved in signaling metabolic regulation. More recent studies on other mechanistic proposals have focused on signaling processes like downregulation of nuclear factor erythroid 2-related factor 2 (Nrf2) signaling, which may be partially responsible for the development of hepatocellular apoptosis in the course of the ROS-dependent liver injury elicited by Cr^6+^. Concomitantly, apoptosis signal-regulated kinase (ASK1)/JNK-signaling activity was upregulated. While Cr^6+^ is highly toxic, Cr^3+^ is appreciated as a nutritional supplement.	Teschke, 2022 [2] DesMarais, 2019 [85]
Copper (Cu)	Copper ions enter the liver via the portal vein following enteral absorption and become part of the hepatocellular homeostasis, where they can produce free radicals, which in turn augments cellular oxidative stress. Via the Haber Weiss and Fenton reactions high levels of free radicals are generated, which attack cellular proteins and phospholipids, disrupting special functional enzymes of the respiratory chain. This process is called cuproptosis and can lead to programed cellular death.	Teschke, 2024 [4] Teschke, 2024 [12] Chen, 2020 [86]
Iron (Fe)	Iron toxicity is attributable to free radical generation facilitated by the Haber Weiss and Fenton reactions involving Fe^2+^ and H_2_O_2_. These processes lead to the catalytic generation of the highly reactive and toxic hydroxyl radicals characterized by a half-life and a reactivity within micro- to nano-second range. Similarly, the formation of other free radical species like superoxide and nitrous oxide, as well as oxygen-activated products such as hydrogen peroxide, is known. They can all be involved in oxidative chain reactions and promote the cascades of the injury, leading to injury of virtually all known organs containing biomolecules such as DNA, structural proteins, sugars, and lipids. The processes start with oxidative stress of mitochondria and the endoplasmic reticulum, where membranes become destructed through lipid peroxidation of poly-unsaturated fatty acids (PUFA). This disruptive process is promoted by ferroptosis and may end in programed cell death.	Teschke, 2024 [4] Teschke, 2024 [11] Kontoghiorghes, 2023 [87]
Lead (Pb)	Lead is known for its toxic properties to virtually all organ systems of the human body. The exact mechanistic background of the mechanistic and molecular steps leading to injury elicited by lead are largely unknown but may occur via cellular oxidative stress. Lead can easily bind to structural proteins with sulfhydryl groups and to cytosolic proteins such as glutathione. Binding causes problems because it lowers the antioxidant defense functions, leading to increased subcellular toxicity due to lipid peroxidation of cell membranes such as mitochondria or the endoplasmic reticulum. Even worse, the high affinity of lead to protein sulfhydryl groups diminishes the activities of a number of enzymes, among which are catalase, glutathione peroxidase, glucose-6-phosphate dehydrogenase, and superoxide dismutase. Genetic variations, known as gene polymorphisms, increase health risks among susceptible persons exposed to lead. In addition, cellular epigenetic regulation governs lead toxicity because Single-Nucleotide Polymorphisms (SNPs) in various genes are associated with the risk of lead poisoning. These genes modify δ-Amino Levulinic Acid Dehydratase (ALAD), Divalent Metal Transporter (DMT’s), Transferrin (TF), Metallothionein (MT), and Vitamin D receptor (VDR).	Teschke, 2022 [2] Mitra, 2019 [88]
Manganese (Mn)	Manganese is involved in developmental disorders due to free radicals in mediating dopaminergic (DAergic) neurodegeneration. At the molecular level, ROS help generate quinines via processes of dopamine autooxidation through redox cycling of Mn^2+^ and Mn^3+^, a reaction leading to ROS and DA-*o*-quinone and resulting in cellular oxidative damage. In essence, high autoxidation rates of cytoplasmic dopamine may contribute to DAergic cell death through the formation of cytotoxic quinones and ROS.	Avila, 2013 [89]
Mercury (Hg)	Methylmercury (MeHg), with its toxic potential, leads to impaired metabolic pathways, affecting cellular and molecular homeostasis in the brain. Cytokines, oxidative stress, mitochondrial malfunction, disturbed Ca^2+^, and disrupted glutamate homeostasis are responsible for brain cell injuries, and all of these contribute to cell death. MeHg is a serious neurotoxin due to its property of rapid passing the blood–brain barrier. The molecular and mechanistic mechanism of toxicity caused by inorganic mercury compounds can be traced back to the disruption of cell membranes by affecting cell functions and impairing cell permeability. The inorganic mercury compounds start binding to thiol groups of structural proteins, leading to molecular dysfunction due to protein denaturation. In addition, the central nervous system may be affected by neuroinflammation.	Rukhan, 2024 [90] Wu, 2024 [91]
Nickel (Ni)	As nickel in drinking water does not cause health problems, mechanistic considerations are not warranted.	Peters, 2022 [76]
Zinc (Zn)	Zinc is detectable in drinking water in only small amounts and not implicated in health issues, making further mechanistic discussions unnecessary.	Huang, 2015 [80]

Abbreviation: ROS, reactive oxygen species.

**Table 3 ijms-26-11656-t003:** Cascade of events triggered by the Haber Weiss and/or Fenton reactions.

Reaction Type	Haber Weiss and Fenton Reactions
Copper-based Haber Weiss reaction Copper-based Fenton reaction	Cu^2+^ + •O_2_^−^ → Cu^1+^ + O_2_ Cu^1+^ + H_2_O_2_ → Cu ^2+^ + OH− + •OH
Iron-based Haber Weiss reaction Iron-based Fenton reaction	Fe^3+^ + •O_2_^−^ → Fe^2+^ + O_2_ Fe^2+^ + H_2_O_2_ → Fe^3+^ + OH− + •OH
Copper-based Net reaction Iron-based Net reaction	•O_2_− + H_2_O_2_ → OH− + •OH + O_2_ •O_2_− + H_2_O_2_ → OH− + •OH + O_2_

This table was taken from an article published in an open access journal [12]. Notably, the so-called Net reaction results from the combined action of the Haber Weiss reaction with the Fenton reaction.

## Data Availability

No new data were created or analyzed in this study.

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
