# Peer review of "Heavy Metals Like Aluminum, Arsenic, Cadmium, Chromium, Copper, Iron, Lead, Manganese, Mercury, Nickel, and Zinc Polluting the Drinking Water: Their Individual Health Hazards"

_ijms, 2025, doi:10.3390/ijms262311656_

Round 1
Reviewer 1 Report
Comments and Suggestions for Authors
In this review, the authors discussed the accumulation of various HMs in drinking water, which can provoke various diseases and pose significant burdens on human health. This review presents valuable information about the critical health risks incited by exposure to HMs, which necessitate proactive management and precise biomonitoring regulations by national and international regulatory authorities. Some comments should be considered before accepting this review for publication in IJMS as follows:
- Abstract: The authors should present the difference between this review and previously published review articles that discussed the same topic.
- Intro: The authors should justify the novelty provided in this review and the aim should be clearly described.
- The review lacks figures to illustrate some topics discussed in the review to attract the readers. Particularly, major information is presented in tables. The authors should include figures showing the toxicity mechanisms of HMs across various human organs. I mean, oxidative stress, inflammation, and cellular apoptosis.
- The authors should outline the accumulation of HMs in the tissues of fish, which could further transfer to humans, compromising the function of human organs.
- The authors should propose the early detection approaches of HMs in drinking water as early warning signals to manage their accumulation. For instance, chemical and biomonitoring evaluations should be highlighted. Please check this article if it could help; https://doi.org/10.3390/antiox13091039.
- The authors should discuss the limitations of HMs control in drinking water and their recommendations accordingly.
Author Response
In this review, the authors discussed the accumulation of various HMs in drinking water, which can provoke various diseases and pose significant burdens on human health. This review presents valuable information about the critical health risks incited by exposure to HMs, which necessitate proactive management and precise biomonitoring regulations by national and international regulatory authorities. Some comments should be considered before accepting this review for publication in IJMS as follows:Conclusion: There is a need to strengthen new policies, encourage innovation, and enhance global collaboration to ensure that orphan drugs continue to transform the management of rare diseases and improve the quality of healthcare and life for affected individuals. Done in new section #9 and at the end of conclusions.
Dear Reviewer 1,
Thank you for giving us the chance to improve our paper and for your valuable proposals, that were considered in the revision:
- Abstract: The authors should present the difference between this review and previously published review articles that discussed the same topic. Done L22-23.
- Intro: The authors should justify the novelty provided in this review and the aim should be clearly described. Done L70-73.
- The review lacks figures to illustrate some topics discussed in the review to attract the readers. Particularly, major information is presented in tables. The authors should include figures showing the toxicity mechanisms of HMs across various human organs. I mean oxidative stress, inflammation, and cellular apoptosis. Good idea but hardly feasible incorporating all HMs and all organs affected. We prefer focusing on the tables that provide clear data.
- The authors should outline the accumulation of HMs in the tissues of fish, which could further transfer to humans, compromising the function of human organs. Done under the section of Minamata disease 6.3, a new ref #115 was added.
- The authors should propose the early detection approaches of HMs in drinking water as early warning signals to manage their accumulation. For instance, chemical and biomonitoring evaluations should be highlighted. Please check this article if it could help; https://doi.org/10.3390/antiox13091039. Sorry, this link was not available for us according to: Doi not found but details are given in new section 7.
- The authors should discuss the limitations of HMs control in drinking water and their recommendations accordingly. Details are given in section #7.
Reviewer 2 Report
Comments and Suggestions for Authors
I found your work very interesting, but I would like to make a few comments/suggestions: This article uses many acronyms, and the full name of the organism, molecule, etc., should always be included. However, this is not always the case: In Table 2, for aluminum, “NF-κB,” “p53,” and “JNK” appear. Next, in the section on cadmium, NF-κB and p53 are explained. This should appear when they are named in aluminum. I have not found the meaning of JNK in the text.
I have also not found the meaning of “Nrf2”, nor of “ASK1-JNK” which appear in table 2 in the chrome section.
The meaning of WHO appears on line 228, when it should be on line 200.
Section 5.2 mentions the consumption of Ipomea aquatica by local villagers. To which country or area does it refer?
The National Institute for Occupational Safety and Health is mentioned on line 225. In which country is it?
What criteria were used to select the references used for this review? In my opinion, a review should include details of how the review was conducted: Where did you search for the references? What keywords have they used? What was the time criterion used?
Table 1 presents the health risks to humans from exposure to heavy metals in drinking water. Did you only find one bibliographic reference for each heavy metal?
In my opinion, 55 bibliographic references are too few for a review. Nine of those 55 references are self-citations. It's a very high number. You should use more bibliographic references if you want to keep those nine self-citations.
Author Response
Dear Reviewer 2,
Thank you for your proposals to improve our paper. Further down are our answers.
Formularbeginn
I found your work very interesting, but I would like to make a few comments/suggestions: This article uses many acronyms, and the full name of the organism, molecule, etc., should always be included. However, this is not always the case: In Table 2, for aluminum, “NF-κB,” “p53,” and “JNK” appear. Next, in the section on cadmium, NF-κB and p53 are explained. This should appear when they are named in aluminum. I have not found the meaning of JNK in the text. I have also not found the meaning of “Nrf2”, nor of “ASK1-JNK” which appear in table 2 in the chrome section. Expansions were included.
The meaning of WHO appears on line 228, when it should be on line 200. Done.
Section 5.2 mentions the consumption of Ipomea aquatica by local villagers. To which country or area does it refer? Done: Vietnam.
The National Institute for Occupational Safety and Health is mentioned on line 225. In which country is it? Done: US
What criteria were used to select the references used for this review? In my opinion, a review should include details of how the review was conducted: Where did you search for the references? What keywords have they used? What was the time criterion used? Done in new para 2. Literature Search.
Table 1 presents the health risks to humans from exposure to heavy metals in drinking water. Did you only find one bibliographic reference for each heavy metal? Thank you, great point, other reports were added.
In my opinion, 55 bibliographic references are too few for a review. Nine of those 55 references are self-citations. It's a very high number. You should use more bibliographic references if you want to keep those nine self-citations. We completely agree. Done, we have now 115 references through adding reports of other authors.
Round 2
Reviewer 1 Report
Comments and Suggestions for Authors
The authors improved the quality of the work based on the revision following initial queries. Therefore, I believe this review is now ready to be published in IJMS.
Author Response
Thank you for posive notes.
Rolf Teschke
Reviewer 2 Report
Comments and Suggestions for Authors
Congratulations on your article. The new version is a significant improvement.
The only thing that needs correcting is the format of the bibliographic references. References 13 through 81, and also 91, are not formatted the same as the rest. Also check the font type and size, and the line spacing
Author Response
Dear Reviewer 2,
Thank you for your comments. I checked the bibliography and could not see any problems that I could improve. If improvement are needed, I guess the editorial processing department will take care of.
Kind regards,
Rolf Teschke